# Virucidal PVP-Copper Salt Composites against Coronavirus Produced by Electrospinning and Electrospraying

**DOI:** 10.3390/polym14194157

**Published:** 2022-10-04

**Authors:** João de Deus Pereira de Moraes Segundo, Jamilly Salustiano Ferreira Constantino, Guilherme Bedeschi Calais, Celso Fidelis de Moura Junior, Maria Oneide Silva de Moraes, Jéssica Heline Lopes da Fonseca, Junko Tsukamoto, Rodolpho Ramilton de Castro Monteiro, Fábia Karine Andrade, Marcos Akira d’Ávila, Clarice Weis Arns, Marisa Masumi Beppu, Rodrigo Silveira Vieira

**Affiliations:** 1Department of Chemical Engineering, Federal University of Ceará, Fortaleza 60455-760, Brazil; 2Department of Materials and Bioprocess Engineering, University of Campinas, Campinas 13083-852, Brazil; 3Department of Manufacturing and Materials Engineering, University of Campinas, Campinas 13083-860, Brazil; 4Thematic Laboratory of Microscopy and Nanotechnology, National Institute of Amazonian Research, Manaus 69067-001, Brazil; 5Department of Genetics, Evolution, Microbiology and Immunology, University of Campinas, Campinas 13083-970, Brazil

**Keywords:** electrospinning technology, polyvinylpyrrolidone, copper, coronavirus

## Abstract

Electrospinning technology was used to produced polyvinylpyrrolidone (PVP)-copper salt composites with structural differences, and their virucidal activity against coronavirus was investigated. The solutions were prepared with 20, 13.3, 10, and 6.6% *w*/*v* PVP containing 3, 1.0, 0.6, and 0.2% *w*/*v* Cu (II), respectively. The rheological properties and electrical conductivity contributing to the formation of the morphologies of the composite materials were observed by scanning electron microscopy (SEM). SEM images revealed the formation of electrospun PVP-copper salt ultrafine composite fibers (0.80 ± 0.35 µm) and electrosprayed PVP-copper salt composite microparticles (1.50 ± 0.70 µm). Energy-dispersive X-ray spectroscopy (EDS) evidenced the incorporation of copper into the produced composite materials. IR spectra confirmed the chemical composition and showed an interaction of Cu (II) ions with oxygen in the PVP resonant ring. Virucidal composite fibers inactivated 99.999% of coronavirus within 5 min of contact time, with moderate cytotoxicity to L929 cells, whereas the virucidal composite microparticles presented with a virucidal efficiency of 99.999% within 1440 min of exposure, with low cytotoxicity to L929 cells (mouse fibroblast). This produced virucidal composite materials have the potential to be applied in respirators, personal protective equipment, self-cleaning surfaces, and to fabric coat personal protective equipment against SARS-CoV-2, viral outbreaks, or pandemics.

## 1. Introduction

COVID-19 is a complex severe acute respiratory syndrome disease caused by the SARS-CoV-2 virus, which is primarily responsible for the current pandemic [1]. In this scenario, the World Health Organization (WHO) has systematically established measures to respond to and cope with the disease [2]. However, despite advances in vaccines, the emergence of new variants continues to concern scientists [3] because these variants undergo genetic mutations in the spike glycoprotein, opening the possibility of preventing the action of neutralizing antibodies [4]. The spike protein is the target of virtually all neutralizing antibodies [5]. Furthermore, these mutations can affect the properties of the virus, making the disease more aggressive and/or increasing the rate of transmissibility [6]. Therefore, prevention should be the first line of action against the coronavirus from an epidemiological point of view [7]. Established prevention measures include the disinfection of environments and surfaces [8].

Given the above information, electrospinning is an emerging and versatile technology capable of producing fibers, spheres, and beads with micro- or nanometric diameters through the application of electrostatic forces on a fluid (i.e., normally polymer solution) [9]. The resulting material is usually collected on a grounded metallic plate and may coat a material placed on it, such as a fabric. The technique setup can vary; fibers may be collected by rotating cylinders, disks, rings, or other grounded materials with varying shapes [10]. The fluid can be expelled from a perpendicular capillary by gravitational forces or by the pressure exerted by a pump if disposed horizontally [11].

Several experimental parameters must be controlled because they significantly affect the product formation, presence and number of defects, and the final material characteristics. Among the fluid parameters that affect the process are polymer choice, concentration, molecular weight distribution, molecular structure, viscosity, conductivity, and surface tension [12]. Processing and ambient parameters that must be controlled include the flow rate, applied voltage, tip-collector distance, temperature, and relative humidity [13]. Polymer parameters may affect the fiber shapes and sizes, such as circular, ribbon, or split fiber shapes, due to the fluid mechanical effects, the solvent evaporation rate, or electrical charging of the fluid. High concentration, surface tension, or viscosity may promote the formation of polymer entanglements or prevent fluid motion, whereas low values of these properties may cause the fibers to collapse into droplets before reaching the collector surface. Processing parameters such as tip-collector distance and electrical field may exert an opposite influence on the bead density. Ambient conditions, such as temperature and humidity, affect the solvent evaporation rate [11]. Frenot and Chronakis reported that controlling these parameters enable control of not only the diameter and internal morphology of the fibers but also their surface structure and functionality, allows for the development of new assembly strategies [11].

The polymer solution concentration is one of the defining parameters with respect to the shape of the fibers, beads, or spheres [9,14,15]. For instance, spheres can be produced by the electrospraying method, which is derived from electrospinning technology and is also promoted by the application of electrostatic forces [16]. Spheres and beads are commonly used in the pharmaceutical [17] and biomedical industries [18].

The considerable versatility of this technique makes it applicable in multiple research fields, such as biomedicine, water filtration, corrosion resistance, photocatalysis, and sensorics, among others. Katti et al. produced electrospun nanofiber systems for wound healing and drug delivery applications. They were able to incorporate the antibiotic cefazolin into a biodegradable polymer solution prior to the electrospinning process, obtaining drug-loaded nanofibers [19]. Zhang et al. produced an electrospun polyimide membrane that was coated with PDMS and ZnO for high-efficiency oily wastewater treatment [20]. Rivero et al. coated an aluminum alloy with polyvinyl chloride (PVC) through electrospinning to produce a superhydrophobic surface for anticorrosive purposes [21] and reviewed other similar studies [22,23]. Albistur et al. developed a multifunctional coating that presented combined anticorrosion and photocatalytic properties [24]. Poly(acrylic acid) (PAA) and *β*-cyclodextrin (*β*-CD) were used to provide the structure of a fiber mat. Titanium and iron compounds were incorporated into a solution by dissolution prior to the electrospinning process. These compounds conferred anticorrosion and photocatalytic properties to the produced material.

Polyvinylpyrrolidone (PVP) is a biodegradable, biocompatible, water-soluble, pH-stable, nontoxic polymer and is widely used to produce nanocapsules, implant material, and scaffolds. It has application potential in novel technologies against coronavirus disease 2019 (COVID-19), such as production of nasal sprays and in situ gel for drug release [15].

Researchers theoretically demonstrated that the release of Cu (II) ions causes the destruction of viral proteins and genomes; in addition, inactivation of the virus by Cu (II) may be related to direct and indirect activity with viral surfaces via production of reactive oxygen species (ROS) [25]. Copper interferes with important virus proteins, causing their inactivation [26]. Cu (II) ions irreversibly inhibit HIV protease, an important protein for viral replication [27].

The production of a filtration system composed of a nanofibrous matrix of polylactic acid and cellulose acetate containing copper oxide nanoparticles and graphene oxide nanosheets by electrospinning provided an innovative solution against SARS-CoV-2 [28]. Nanofibrous membranes produced by electrospinning based on polymers, such as polyacrylonitrile (PAN) with plant extracts of Ag/g-C_3_N_4_ (Ag-CN), Myoporum bontioides, and silver nanoparticles, significantly inhibited influenza A virus (H3N2) and exhibited a bactericidal effect of more than 98.65 ± 1.49% and 97.8 ± 1.27% against *E. coli* and *S. aureus* [29]. Studies report the virucidal and antimicrobial efficiency of nanofibrous materials produced by electrospinning, which are important allies in terms of conferring antiviral properties to masks and personal protection equipment (PPE) and improving their ability to prevent microbial diseases [29,30,31].

Recently, the cold-spray technique was used for the rapid coating of copper on steel parts, and a virucidal assay showed that copper was able to inactivate Sars-CoV-2 [32]. In this context, copper derivatives represent alternatives to produce new materials, such as protective clothing, PPE, and other essential virucidal surfaces, and to establish methods that can be used and explored to prevent Sars-CoV-2 and its spread. Thus, our research shows the versatility of electrospinning technology in the production of microspheres and microfibers of PVP containing copper sulfate.

In the present study, virucidal composite fibers and microparticles were produced by the electrospinning technique, using copper salt as a bioactive compound against coronavirus. The properties of the electrospinning solutions, such as viscosity and electrical conductivity, were measured to improve the processing of electrospun and electrosprayed composites. The virucidal copper-based composites were characterized using scanning electron microscopy (SEM), energy-dispersive X-ray spectroscopy (EDS), and Fourier transform infrared spectroscopy (FTIR). A virucidal assay was performed by exposition to the MHV-3 Coronavirinae lineage within 24 h. In addition, a cytotoxicity assay on L929 cells.

## 2. Materials and Methods

### 2.1. Materials

PVP (MM = 1,300,000 g/mol) and copper sulfate (MM = 249.69 g/mol) were purchased from Sigma-Aldrich, and methyl alcohol from was purchased from Neon Comercial Ltda. Additionally, deionized water with an electrical conductivity of 0.5 μS/cm was used in this study.

### 2.2. Preparation of the Solutions

First, PVP was dissolved using water:methanol mixtures at a 2:1 volume ratio, which was fixed under mechanical stirring for 2 h to prepare all solutions used in this study. Then, several copper salt concentrations were added to the PVP solution and vigorously stirred for 24 h without heating. Control samples were prepared similarly but without the addition of copper salt. Table 1 lists the concentration of PVP and copper salt used to prepare the solutions, their solution properties, needle type, and the coding of each solution.

### 2.3. Electroctrospinning and Electrospraying of the Solutions

In terms of electrospinning technology, both methods used a high-precision infusion pump, a high-voltage source (up to 30 kV), a metallic needle, and a static rectangular collector. Moreover, the positive electrode of the high-voltage source was connected to the metallic needle, and its negative electrode was connected to the metallic collector.

The parameters of the electrospinning method were: needle diameter, Ø_Needle_ = 21G; flow rate, Q = 4 mL/h; working distance, Wd = 17 cm; and applied tension, V = 14 kV. The following parameters were applied for the electrospraying method: Ø_Needle_ = 27G; Q = 2 mL/h; Wd = 10 cm; and V: 10 to 14 kV. The environmental parameters registered during the production of the PVP-copper salt composites were a temperature of approximately 21 °C and a relative humidity of 54%. Figure 1 shows a schematic summary of the methodology used to prepare the PVP-copper salt composites.

### 2.4. Electrical Conductivity

The electrical conductivity of the solutions was measured using a conductivity meter (Tecnal, TEC-4MP model) equipped with a stainless steel temperature sensor and a glass cell (K = 1). The measurements were conducted in duplicate at 25 °C, and the results were represented in µS cm^−1^.

### 2.5. Rheology

Rheological characterization was performed using a rheometer (MCR-102, Anton Paar, Graz, Australia) with a cone plate geometry of 50 mm diameter and a 0.9815° cone angle at 25 °C. Steady-state viscosity tests were performed in the 0.1 to 600 s^−1^ range. Frequency sweep tests were performed within the linear viscoelasticity region (LVE) with an amplitude of 1% in the angular frequency range of 1 to 400 rad/s. The shear rates of the needle of 21G and 27G were calculated based on the literature [33], and the viscosity was obtained as a function of the shear rate.

### 2.6. Morphological Characterization and Elemental Analysis

Scanning electron microscopy (SEM, VEGA3,TESCAN, Brno, Czech Republic) was performed to observe the morphology of the electrospun and electrosprayed solutions. The samples were previously coated with gold sputtering with ca. 10 nm thickness (BAL TEC, CPD 050, Balzers Union, Balzers, Liechtenstein). Energy-dispersive X-ray spectroscopy (EDS) (Oxford AZtecO 4.3 software) was used to investigate the presence of copper in the composition of the electrospun and electrosprayed solutions.

### 2.7. Composite Fiber and Microparticle Diameter Measurements

Diameter measurements of composite fibers and microparticles of the virucidal composites were measured from SEM images using ImageJ software [34]. Diameter values were expressed as average diameter ± standard deviation obtained by manually measuring 100 fibers per SEM image for each sample.

### 2.8. Chemical Characterization by Infrared Spectroscopy

Chemical characterization by infrared spectroscopy using a Nicolet iS5 FT-IR spectrometer (Thermo Scientific, Waltham, MA, US) was performed in the 400 to 4000 cm^−1^ range with a 2 cm^−1^ resolution and 32 scans. All the samples were mixed with potassium bromide (KBr) in a ratio of 1:100 and analyzed at 25 °C.

### 2.9. Virucidal Test and Cytotoxicity Assay

Murine fibroblast cell line L929 cells were grown in Dulbecco’s Minimum Eagle Essential Medium (DMEM) supplemented with 10% fetal bovine serum (FBS) for antiviral activity and cytotoxicity assays. Virucidal properties of the samples were determined in vitro by coronavirus exposure of the MHV-3 *Coronavirinae lineage*, which belongs to the *betacoronavirus* (*β-CoV*) genera, a surrogate of SARS-CoV-1 and SARS-CoV-2 [35,36].

The material strips were exposed to 500 µL of the virus (concentration of 10^8,0^/mL) at room temperature for varying contact times (5, 15, 30, 60, 120, and 1440 min), followed by serial dilution from 10^1^ to 10^8^ to a monolayer of L929 mouse fibroblasts (ATCC^®^ CCL-1™).

After the system was incubated for 48 h in a 5% CO_2_ atmosphere at 37 °C, cytopathic effect (CPE) and cytotoxicity effects were observed in quadruplicate in an inverted microscope. The infectivity rate was calculated based on the method described by Reed and Muench (1938) [37], the log viral reduction (L) was calculated according to Equation (1), and the inhibition efficacy was expressed as percentage reduction (P) using the Equation (2):(1)$$Log\;reduction=log_{10}\left(\frac{A}{B}\right),$$
(2)$$P=\left(1-10^{-Log\;reduction}\right)\times 100\;\left(\%\right)$$
where *A* is the number of viable viruses before treatment (from the initial titer of the viral solution), and *B* is the number of viable viruses after treatment (from the titer of the treated solution).

## 3. Results and Discussion

### 3.1. SEM Morphology, EDS Spectroscopy and Elemental Mapping, and Chemical Composition of the Electrospun and Electrosprayed PVP-Cu (II) Composites

SEM and EDS images of the electrospinning and electrospraying solutions listed in Table 1 are shown in Figure 2 and Figure 3. The optimal electrospinning conditions were previously determined to produce composite fibers free of beads, as shown in Figure 2a. SEM images show the predominance of the random fiber morphology, with diameters of 0.80 ± 0.35 µm. Therefore, the formulation of the spun A solution (see Table 1) and the established process parameters are very promising to produce PVP-Cu (II) ultrafine composite fibers via electrospinning.

As shown in Table 1, the PVP and Cu (II) concentrations were reduced from 20 to 6.6% (*w*/*v*) and from 3 to 0.2% (*w*/*v*), respectively, to reduce the viscosity and promote the particle formation by electrospraying process [38,39], as well as to avoid the formation of copper salt precipitates. Moreover, the sprayed A, sprayed B, and sprayed C solutions were electrosprayed with a higher shear rate of ca. 440 s^−1^ using a 27G needle. Figure 2b–d shows the morphologies of the electrospray solutions, highlighting that the sprayed C solution (Figure 2d) successfully resulted in PVP-Cu (II) composite microparticles with diameters of 1.50 ± 0.70 µm. Thus, the morphologies were clearly affected by concentration changes of the electrospray solutions, as shown more clearly in Figure 3. Figure 4 shows the result with respect to the size distribution of the produced microparticles and fibers.

Figure 5a–d displays the EDS spectroscopic images of the spun A, sprayed A, sprayed B, and sprayed C solutions after processing by electrospinning and electrospraying. The peaks that identify copper in the EDS spectrum, i.e., Cu at 0.928 keV and at 8.040 keV [40], decrease in intensity as the copper salt concentration decreases, corroborating the atomic percentage (at%) of copper of 0.93, 0.54, 0.26, and 0.2 at% in electrospray spun A, sprayed A, sprayed B, and sprayed C solutions, respectively, and confirming copper incorporation in all cases.

### 3.2. Investigation of the Formation of PVP-Cu (II) Composite Fibers and Microparticles by Rheological Study and Electrical Conductivity

Results of shear viscosity as a function of shear rate are presented in Figure 6a and listed in Table 1. The spun A solution showed an initial Newtonian plateau followed by gradual shear thinning. Sprayed A, sprayed B, and sprayed C solutions did not exhibit Newtonian behavior, but their viscosity slightly decreased with increased shear rate. Viscoelastic characteristics analysis showed an elastic property (G’) in spun A and sprayed A solutions (Figure 6b). Moreover, we observed an increase in G’ in the spun A solution compared the G’ of spun A without copper salt, which may be attributed to interactions of the Cu^2+^ ions with the PVP chains. Figure 6c shows the predominance of G’’ at lower frequencies, whereas at higher frequencies, G’ predominates. A crossover of G’ and G’’ occurs at 183 rad s^−1^ for spun A, whereas for sprayed A, the crossover occurs at 125 rad s^−1^ (Figure 6d). Table 1 presents a summary of the values of the electrical conductivity, which decreased with decreased copper salt content and were considerably higher compared to those of PVP solutions, all of which measured 13.5 µS cm^−1^.

### 3.3. Chemical Composition by IR Spectroscopy and Study of the Cu^2+^ Interaction with PVP

IR spectroscopy was used to confirm the chemical composition of various PVP-copper salt composites and to identify the vibrational modes present in the PVP affected by increased copper salt concentration. Figure 7a shows the vibrational bands of the pure PVP and copper salt spectra according to the literature [41,42,43,44].

Despite the varying Cu (II) concentrations, the FTIR spectra of the electrospraying spun A, sprayed A, sprayed B, and sprayed C solutions are strongly represented by an original spectral shape of the PVP, as shown in Figure 7a. The C=O group changed in intensity with the addition of copper salt, but this change was not proportional to the increase in its concentration. On the other hand, we observed a transmittance relationship between the intensity of the C=O group (1650 cm^−1^) and that of the OH group (3450 cm^−1^).

Cu^2+^ ions are transition metals with chelating properties due to empty *d* orbitals that enable them to make coordination connections with atoms with isolated pairs, such as oxygen (O), nitrogen (N), and fluorine (F), among others [45]. We observed an increase in peak intensity at 1120 cm^−1^ with increasing copper salt concentration (see Figure 7b). This result suggests an interaction of Cu^2+^ ions with oxygen in the PVP resonant ring (C=O-Cu), as the metal ions can easily bond with C=O and C-OH groups [45,46]. The molecular interaction between Cu^2+^ and PVP is represented in the schematic illustration shown in Figure 7c.

### 3.4. Virucidal Test and Cytotoxicity Assay of the PVP-Copper Salt Composites

The virucidal efficiency of the PVP-copper salt composite fibers and microparticles corresponding to spun A and sprayed C solutions produced by electrospinning and electrospraying, respectively, was evaluated using a viral model that belongs to the same family as SARS-CoV-2, the MHV-3 coronavirus. As shown in Table 2, the PVP-copper salt composite fibers exhibited excellent virucidal efficiency of 99.999% within 5 min of contact time, maintaining this antiviral effect for as long as 1440 min, indicating immediate virucidal activity against the coronavirus.

As shown in Table 2, the virucidal action behavior of the PVP-copper salt composite microparticles varied in a time-dependent manner with respect to virucidal efficiency; after 30 min of contact time, no effect against coronavirus was observed, whereas after 60 min of contact time, the virucidal efficiency was 99% and 99.999% after 1440 min of contact time.

Manakhov et al. [47] reported that the increased antiviral activity of copper-containing nanofibers was the result of the release of ions that offered high antiviral activity against the SARS-CoV-2 virus strain. Thus, our study results suggest that this virucidal action occurs as a result of the release of Cu^2+^ ions from copper sulfate dissociation into the culture medium of the virucidal test, which can promote an electrostatic interaction of Cu^2+^ ions with the viral model, leading to the destruction of the coronavirus. Moreover, the interaction of Cu^2+^ ions with coronavirus RNA can promote ROS production and cause its destruction [48,49].

In the cytotoxicity assay, the control samples exhibited a slight level of virucidal activity against coronavirus with a low toxicity effect, which might have been caused by solvent retention of the residual methanol that remained in controls after their production by electrospinning and electrospraying [50,51,52].

The PVP-copper salt composite fibers exhibited a moderate toxicity effect against L929 cells after 1440 min of exposure, whereas a low toxicity effect was observed for after the same exposure time for PVP-copper salt composite microparticles, as shown in Table 2. Therefore, increasing the copper salt content considerably improves the virucidal action of the composite against coronavirus but increases the toxicity effect. 

We speculate that there may be an optimal concentration of copper salt that can offer a time-dependent virucidal effect with no toxicity effect to the composites produced by electrospinning or electrospraying, as a high concentration of copper is already expected to have a toxic effect [53].

## 4. Conclusions

In this work, we reported the manufacture of electrospun and electrosprayed PVP-copper salt composites with virucidal properties an investigated the effect of varying concentrations of PVP and copper salt, as well as that of high shear rate, on the electrospinning process. EDS spectroscopy showed the incorporation of copper, whereas SEM images confirmed the structural differences of the virucidal composites. The different electrospinning solutions presented changes in their rheological properties and electrical conductivity. Electrospinning produced PVP-copper salt composite fibers due to the high viscosity and good elastic properties (G’) of the spun A solution. Decreasing the viscosity and applying a high shear rate by needle promoted the formation of the electrosprayed PVP-copper salt composite microparticles. The IR spectra showed a chemical interaction of the Cu^2+^ ions with oxygen in the PVP resonant ring. The virucidal ultrafine composite fibers effectively inactivated coronavirus by 99.999% after 5 min of exposure, exhibiting moderate cytotoxicity against L929 cells, and the virucidal composite microparticles inactivated 99.999% of coronavirus after 1440 min of exposure with low cytotoxicity to L929. Thus, virucidal ultrafine composite fibers have the potential to be used in air filtration applications against SARS-CoV-2, such as in respirators and personal protective equipment, whereas virucidal composite microparticles can be applied to self-cleaning surfaces and fabric coating of personal protective equipment against SARS-CoV-2.

## Figures and Tables

**Figure 1 polymers-14-04157-f001:**
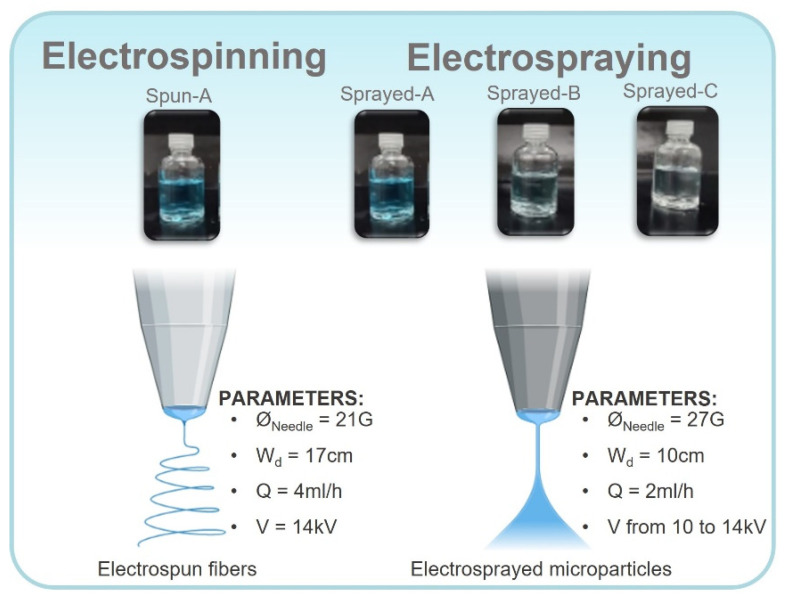
Methodological scheme showing the prepared PVP-copper salt solutions and the electrospinning and electrospraying parameters. Adapted from “Electrospinning instrumental” and “Electrospray emitter” by BioRender.com (accessed on 13 September 2022).

**Figure 2 polymers-14-04157-f002:**
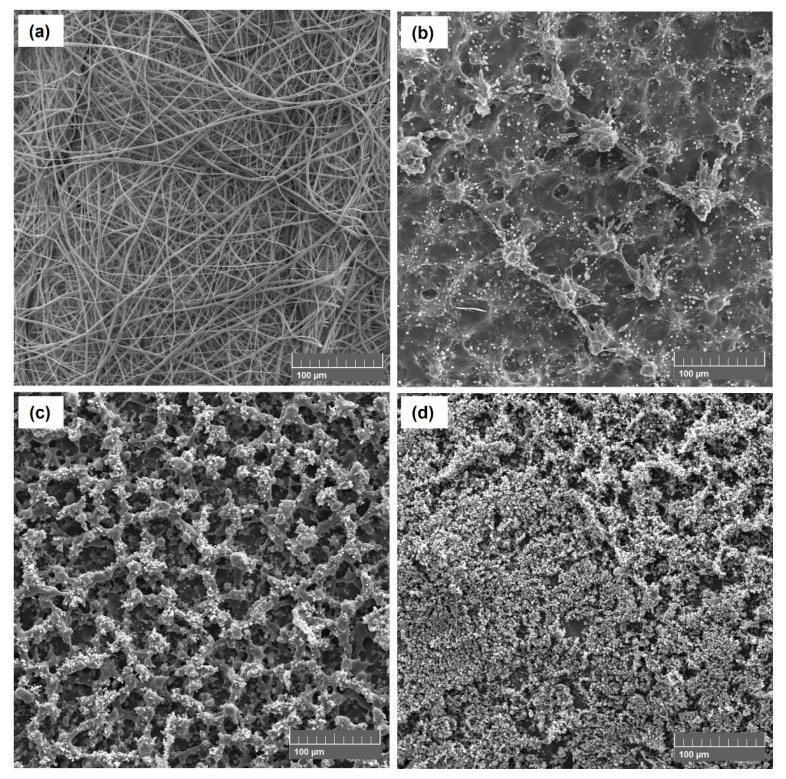
SEM images of spun A (**a**), sprayed A (**b**), sprayed B (**c**), and sprayed C solutions (**d**). Scale bar = 100 µm.

**Figure 3 polymers-14-04157-f003:**
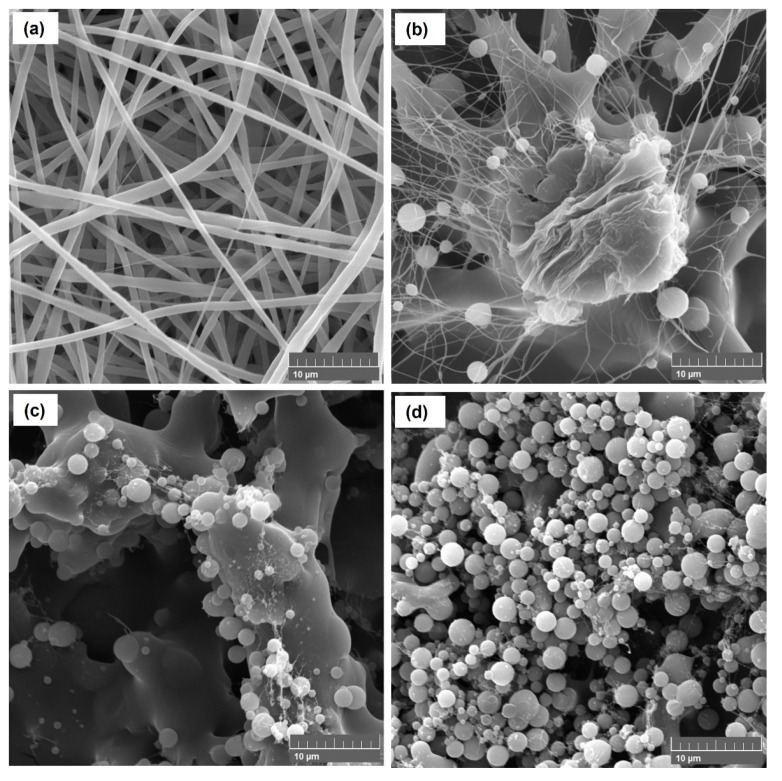
SEM images of the spun A (**a**), sprayed A (**b**), sprayed B (**c**), and sprayed C solutions (**d**). Scale bar = 10 µm.

**Figure 4 polymers-14-04157-f004:**
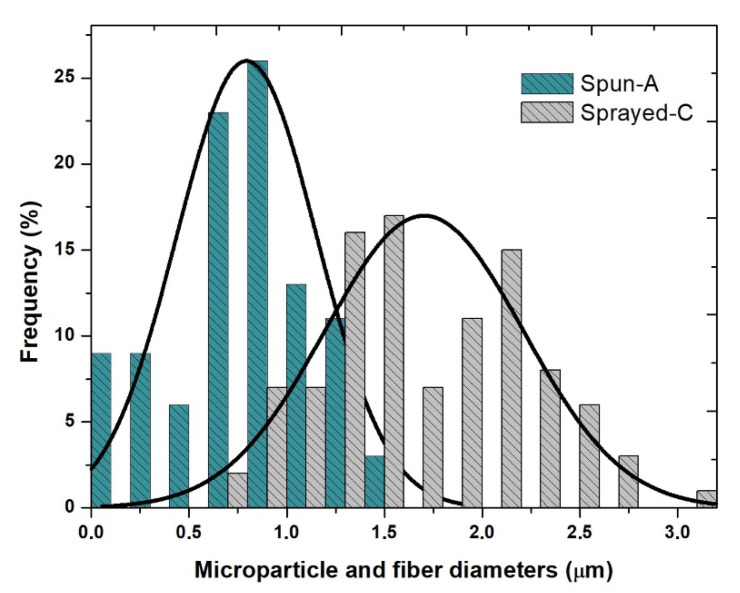
Histogram of size distribution of the PVP-copper salt composite fibers (spun A) and PVP-copper salt composite microparticles (sprayed C).

**Figure 5 polymers-14-04157-f005:**
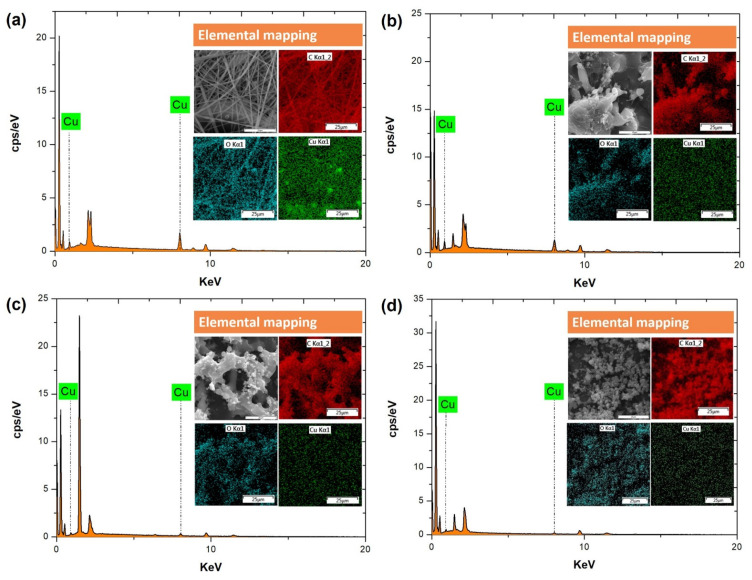
Elemental mapping and EDS spectra after the processing of the spun A (**a**), sprayed A (**b**), sprayed B (**c**), and sprayed C solutions (**d**).

**Figure 6 polymers-14-04157-f006:**
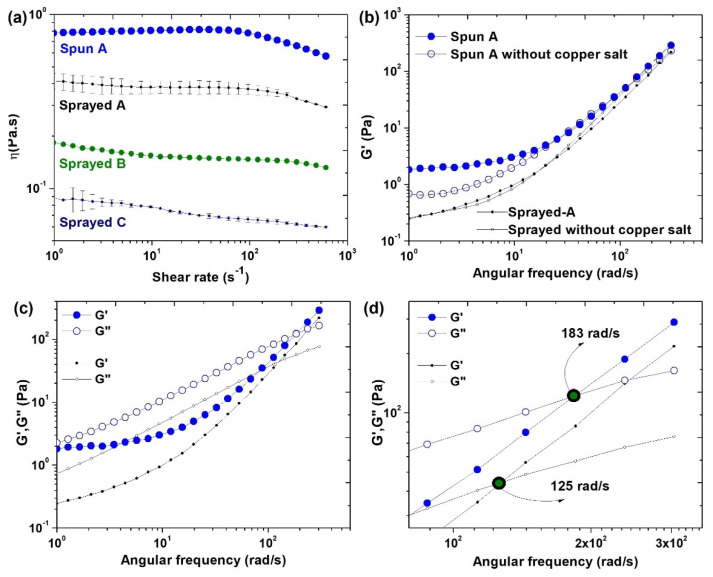
Rheological properties of the electrospinning and electrospraying solutions. (**a**) Steady-state viscosity curves of the spun A, sprayed A, sprayed B, and sprayed C solutions; (**b**) dependence of G’ on angular frequency; (**c**) dependence of G’ and G’’ as a function of the angular frequency; and (**d**) crossover G’ and G’’ for spun A and sprayed A. Note: sprayed B and sprayed C did not show cross-over.

**Figure 7 polymers-14-04157-f007:**
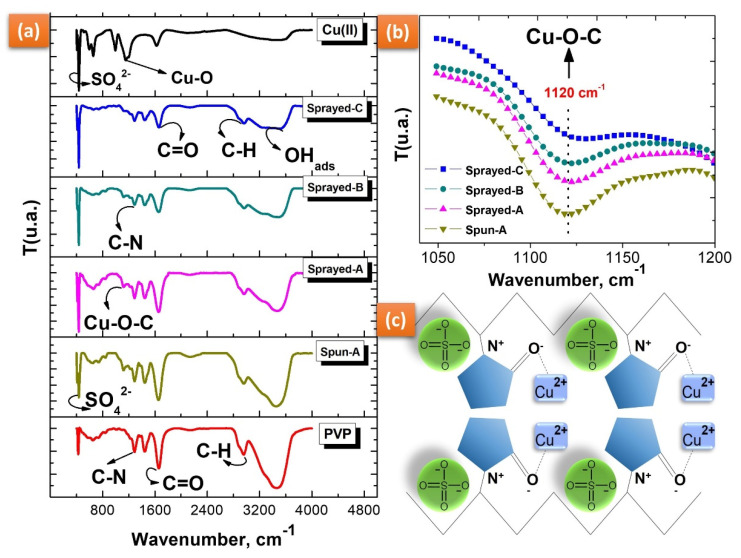
FTIR spectra of (**a**) pure copper salt (Cu (II)), pure PVP, spun A, sprayed A, sprayed B, and sprayed C after processing by electrospinning and electrospraying. FTIR spectra region from (**b**) 1050 to 1200 cm^−1^ (**b**) and (**c**) schematic representation showing the interaction between Cu^2+^ and PVP.

**Table 1 polymers-14-04157-t001:** Formulation concentrations, needles, electrical conductivity, rheological properties, and coding of electrospinning and electrospraying solutions.

PVP %(*w*/*v*)	Copper Salt %(*w*/*v*)	Needle	σ * (µS/cm)	Viscosity (Pa.s)	Crossover (rad/s)	Code
20	3	21G	1361	0.80	183	Spun A
13.3	1	27G	872	0.40	125	Sprayed A
10	0.6	27G	626	0.20	No	Sprayed B
6.6	0.2	27G	350	0.08	No	Sprayed C

* Electrical conductivity of the solutions.

**Table 2 polymers-14-04157-t002:** Results of the coronavirus virucidal test and cytotoxicity assay of the virucidal PVP-copper salt composites and their respective control samples.

Sample	Contact Time (Min.)	Coronavirus (%) ^c^	L929 Cells
Spun A/Control ^a^	5	No affect	Low toxicity
	30	No affect	
	60	99 ^d^	
	1440	99	
PVP-copper salt composite fibers(Spun A)	5	99.999	Moderate toxicity
	30	99.999	
	60	99.999	
	1440	99.999	
Sprayed-C/Control ^b^	5	No affect	Low toxicity
	30	No affect	
	60	99	
	1440	99	
PVP-copper salt composite microparticles (Sprayed-C)	5	No effect	Low toxicity
	30	No effect	
	60	99	
	1440	99.999	

^a^ Spun A/Control = spun A without copper salt, ^b^ Sprayed-C/Control = sprayed C without copper salt, ^c^ inhibition efficiency (%), ^d^ not considered virucidal fiber 99% < 99.99%.

## Data Availability

Not applicable.

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
