# Peer review of "Virucidal PVP-Copper Salt Composites against Coronavirus Produced by Electrospinning and Electrospraying"

_polymers, 2022, doi:10.3390/polym14194157_

Round 1
Reviewer 1 Report
It is not stated which source was used for + or -. The distance between the needle and the collector is not indicated - due to the electrical intensity.
It would be appropriate to measure the breathability of the materials.
Reviewer 2 Report
In this work, it is presented a novel and interesting study about the optimization of virucidal PVP-copper salt composites by using the combination of two different wet-chemistry fabrication techniques known as the electrospinning and electrospraying, respectively, as a function of the experimental deposition parameters by varying the resultant applied voltage. In the opinion of this reviewer, the length of the article is not enough for a manuscript published in a scientific journal by showing only three Figures and two Tables, and due to this, it should be considered as a letter. However, a very good explanation in the overall manuscript which it has been supported with SEM images and their corresponding EDS spectroscopy to show the presence of the copper salts inside the polymeric electrospun or electrosprayed coatings as well as the FTIR spectra are of great interest and can facilitate their corresponding understanding for the reader. Finally, I think that this paper can be considered as a publication in Polymers after the following major revisions.
11. There are enough grammatical mistakes along the whole manuscript. Please, revise carefully previous publication. Some examples can be found in the same Abstract or in the Introduction Section (i.e. line 18: structure different; line 22: reveling; line 78: viricidal), among others.
22. In the Introduction section should be larger. In this sense, according to the electrospinning technique, more references related to the variable experimental parameters (flow rate, applied voltage, polymeric concentration, distance tip-collector, relative humidity, among others) should be explained in the manuscript. In addition, te great versatility of this technique makes possible its application in different research fields such as biomedicine, water filtration, corrosion resistance, photocatalysis, sensorics, among others.
Several examples can be incorporated related to this topic and considerably improve the quality of the paper:
“Polymer nanofibers assembled by electrospinning”
“Designing Multifunctional Protective PVC Electrospun Fibers with Tunable Properties”
“Hydrothermal synthesized UV-resistance and transparent coating composited superoloephilic electrospun membrane for high efficiency oily wastewater treatment”
“Electrospinning: A Powerful Tool to Improve the Corrosion Resistance of Metallic Surfaces Using Nanofibrous Coatings”
“Electrospinning of a Functional Perfluorinated Block Copolymer as a Powerful Route for Imparting Superhydrophobicity and Corrosion Resistance to Aluminum Substrates”
“Bioresorbable nanofiber-based systems for wound healing and drug delivery: Optimization of fabrication parameters”
“Evaluation of the Photocatalytic Activity and Anticorrosion Performance of Electrospun Fibers Doped with Metallic Oxides”
3. In order to increase the number of figures in the manuscript, an initial Figure 1 which summarizes the whole manuscript can be a good option to give attention previous the experimental results.
4. In order to have a better morphological characterization, new AFM images can be incorporated in order to have a better appreciation of the resultant fiber diameter and even in the microparticles size distribution. In addition, the wettability of the surfaces can be also measured by water contact angle in order to appreciate any significant differences between both electrospun fibers and electrosprayed particles.
5. Finally, in order to evaluate the mechanical resistance and adhesion of the fibers to the substrate can be considered for long-term applications. In this sense, an easy and rapid test to analyse the resultant coating adhesion should be incorporated in the final manuscript according to ASTM D3359 as a function of the area removed by the tape.
Reviewer 3 Report
This work dealing with interesting electrospun nanocomposites with an efficient antiviral activity. The paper is well written but some improvements are needed:
1) The methodology for measuring the virucidal activity. Why you have chosen this particular method, MHV?
2) Please comment on the Cu ion release. Do you have the kinetics data?
3) Would the toxicity be a problem for respirators? Can be the Cu nanoparticles be detached and inhaled?
4) Some recent works can be referenced and compared to your particular nanofibers. For example: https://www.mdpi.com/1420-3049/27/4/1333
Minor comments:
Line189 ...corroborating with the atomic percentage (wt%) of copper that was 0.93, 0.54, 0.26, and 0.2 wt%
Please either report atomic% or change the wording, because atomic and weight percentage are different.
Please include the copper percentage for the data dealing with the efficiencies.
Round 2
Reviewer 2 Report
The most of the suggestions have been incorporated in this new version. Accept in the present form.